# Evaluation of a Novel DNA Vaccine Double Encoding Somatostatin and Cortistatin for Promoting the Growth of Mice

**DOI:** 10.3390/ani12121490

**Published:** 2022-06-08

**Authors:** Xuan Luo, Zhuoxin Zu, Hasan Riaz, Xingang Dan, Xue Yu, Shuanghang Liu, Aizhen Guo, Yilin Wen, Aixin Liang, Liguo Yang

**Affiliations:** 1Key Laboratory of Agricultural Animal Genetics, Breeding and Reproduction of Ministry of Education, College of Animal Science and Technology, Huazhong Agricultural University, Wuhan 430070, China; xuanxuanzi87@163.com (X.L.); zuzhuoxin@novogene.com (Z.Z.); danxingang2013@163.com (X.D.); shuanghang_liu@webmail.hzau.edu.cn (S.L.); 2Hunan Institute of Animal and Veterinary Science, Changsha 410131, China; 3Department of Biosciences, COMSATS University, Sahiwal Campus, Islamabad 57000, Pakistan; hasan@ciitsahiwal.edu.pk; 4Shandong Provincial Key Laboratory of Biophysics, Institute of Biophysics, Dezhou University, Dezhou 253023, China; yuxuefish@163.com; 5State Key Laboratory of Agricultural Microbiology, College of Veterinary Medicine, Huazhong Agricultural University, Wuhan 430070, China; aizhen@mail.hzau.edu.cn; 6Yongzhou Vocational Technical College, Yongzhou 425100, China; penguinazy@163.com; 7National Center for International Research on Animal Genetics, Breeding and Reproduction (NCIRAGBR), Wuhan 430070, China

**Keywords:** somatostatin, cortistatin, dual expression, growth, DNA vaccine

## Abstract

**Simple Summary:**

Growth trait is one of the most important economic traits for meat animals. Somatostatin DNA vaccine has been proven to enhance the growth rate of animals. However, the growth-promoting effect is not ideal. This study aimed to evaluate the immune effects of a novel eukaryotic dual expression vaccine known as pIRES-S/CST14-S/2SS in mice. Firstly, we demonstrated pIRES-S/CST14-S/2SS could functionally express in GH3 pituitary cells by regulating the growth hormone (GH) and prolactin (PRL) productions. Secondly, we observed that all concentrations of pIRES-S/CST14-S/2SS vaccine could evoke anti-somatostatin (SS) antibodies, leading to a higher level of GH concentration. Notably, pIRES-S/CST14-S/2SS (10 μg/100 μL) immunized mice obtained maximum body weight gain in a booster vaccination period. Our results are helpful for better understanding of the relationship between SS and CST, and pIRES-S/CST14-S/2SS vaccine may be used as a promising alternative for developing growth-promoting vaccine.

**Abstract:**

Animal growth traits are directly linked with the economics of livestock species. A somatostatin DNA vaccine has been developed to improve the growth of animals. However, the growth-promoting effect is still unsatisfying. The current study aimed to evaluate the effect of a novel eukaryotic dual expression vaccine known as pIRES-S/CST14-S/2SS, which encodes the genes obtained by fusing somatostatin (SS) and cortistatin (CST) into hepatitis B surface antigen (HBsAg). After transfection into GH3 cells with pIRES-S/CST14-S/2SS, green fluorescence signals were observed by fluorescence microscopy, suggesting the effective expression of CST and SS in GH3 cells using the IRES elements. Subsequently, both GH and PRL levels were found to be significantly lower in pIRES-S/CST14-S/2SS-treated cells as compared to the control group (*p* < 0.05). Furthermore, the antibody level, hormone secretion, and weight gain in the mice injected with novel recombinant plasmids were also evaluated. The anti-SS antibodies were detectable in all vaccine treated groups, resulting in significantly higher levels of GH secretion (*p* < 0.05). It is worth mentioning that pIRES-S/CST14-S/2SS (10 μg/100 μL) vaccinated mice exhibited a higher body weight gain in the second immunization period. This study increases the understanding of the relationship between somatostatin and cortistatin, and may help to develop an effective growth-promoting DNA vaccine in animals.

## 1. Introduction

Somatostatin (SS) consists of 14 peptides and is a pleiotropic neuropeptide widely distributed throughout the organism, which is generally viewed as a crucial regulator of several hormones, including growth hormone (GH) [1,2], prolactin (PRL) [3], follicular stimulating hormone (FSH) [4], and luteinizing hormone (LH) [5]. Cumulative studies have demonstrated that both active and passive immunization against somatostatin (SS) could induce a rise in growth hormone, thereby improving the growth or lactation performance of animals such as mice [6,7], chickens [8], Hu lambs [9], goats [10], and piglets [11]. However, the growth-promoting effect of animals is still not satisfying, especially for large animals. After the active immunization of SS, the rise of GH hormone is limited, suggesting that there may be another compensatory hormone that affects the secretion of GH.

Cortistatin (CST) is a neuropeptide that displays a high structural and functional homology with somatostatin [12]. Cortistatin can bind to the five native somatostatin receptor subtypes (SSTR1-5) with high affinity [13]. Interestingly, only cortistatin is able to bind other non-SS receptors, such as the ghrelin receptor 1a (GHS-R1a) and Mas-related gene 2 receptor (MrgX2) [14], suggesting it has the ability to trigger unique endocrine or non-endocrine actions than SS. In knockout (KO)-model mice, the absence of endogenous cortistatin(CST^−/−^ mice) provokes a higher GH plasma level compared to normal mice [15]. Importantly, cortistatin levels are increased in the hippocampus of somatostatin knockout mice, indicating CST plays a compensatory effect on somatostatin [16]. This result is subsequently supported by a double-SST/CST-KO mouse model, which exhibits a similar drastic increase in mean circulating GH levels [17]. The in-vitro analysis also showed a biphasic dose-dependent effect of cortistatin on GH secretion from pituitary cells [18]. Those studies reveal that the regulatory effect of cortistatin on the GH production basically match the effect of somatostatin.

Based on the above-mentioned information, we hypothesize that cortistatin may play a compensatory role in the process of in-vivo SS active immunization. We therefore constructed somatostatin and cortistatin eukaryotic dual expression plasmids using pIRES elements, which contain two multiple cloning sites (MCS) and an internal ribosome entry site (IRES), allowing for the high-level expression of two genes. Moreover, the biological functions of both SS and CST were detected in GH3 pituitary cells, and the growth-promoting effects were evaluated in mice.

## 2. Materials and Methods

### 2.1. Plasmids and Gene Synthesis

The recombinant plasmid pCS/2SS encoding S/2SS gene obtained by fusing two copies of somatostatin with hepatitis B surface antigen (HBsAg) was constructed and described in the previous paper [7]. Notably, because both SS and CST belong to small molecular peptides, hepatitis B surface antigen (HBsAg) was used as a carrier molecule to enhance SS and CST immunogenicity. Corstitatin-14 (CST-14, GenBank ID: NM_007745.3) was chemically synthesized and fused (S/CST14) into the 3′ end of the hepatitis B surface antigen (HBsAg) by Sangon Biotech Company (Shanghai, China). Subsequently, two restriction sites, *Nhe* I and *EcoR*I, were fused into the 5′ and 3′ ends of S/CST14, respectively. The eukaryotic expression vector pIRES with double MCS was kindly provided by Dr. Qiuyang Li from Jinan University.

### 2.2. Construction of pIRES-S/CST14-S/2SS Plasmid

The schematic diagram for the construction of the recombinant co-expression plasmid pIRES-S/CST14-S/2SS is presented in Figure 1. Briefly, S/2SS fragment was obtained through the amplification of plasmid pCS/2SS by PCR. An upstream primer with *Sal* I site (underline) is 5′-GCGTCGACCTTCTGAGATGAGTTTTTGTTC

TAC-3′, and a downstream primer with a *Not*I site (underline) is 5′-ATAAGAATGCGGCCGCAATTCCCTGCGCTGAACATG-3′. The amplification was carried out by Taq DNA polymerase (TAKARA, Dalian, China) according to the following procedures: pre-denaturation at 94 °C for 4 min and then 35 cycles of denaturation at 94 °C with each cycle lasting 30 s; annealing at 68 °C for 30 s; extension at 72 °C for 1 min; and final extension at 72 °C for 10 min. After PCR amplification, the fragment was excised at the *Sal*I/*Not*I sites, purified by gel electrophoresis, and then ligated into the pIRES vector to yield the pIRES-S/2SS plasmid. Similarly, the S/CST14 fragment was excised at the *Nhe* I/*EcoR* I sites, purified by gel electrophoresis, and then ligated into the pIRES vector and the pIRES-S/2SS plasmid to generate the pIRES-S/CST14 and pIRES-S/CST14 -S/2SS plasmids, respectively. The plasmids were then transformed into DH5α and preserved in the China Center for Type Culture Collection (CCTCC).

### 2.3. Cell Culture and Transfection

GH3 pituitary cells were purchased from the National Platform of Experimental Cell Resources (China) and used to detect the effect of the overexpression of SS and CST *in vitro* on hormone secretion. GH3 cells were cultured in Ham’s F10 Nutrient Mixture (F10, Hyclone, Hyclone, Logan, UT, USA) supplemented with 2.5% fetal bovine serum (FBS, Gibco, Waltham, MA, USA) and 15% horse serum (HS, Hyclone, Logan, UT, USA). All cells were cultured at 37 °C in an incubator under a humidified atmosphere containing 5% CO_2_. For transfection experiments, the cells were grown until they were a 70–80% confluent monolayer.

The plasmids pIRES-S/CST14-S/2SS, pIRES-S/CST14, pIRES-S/2SS, and pIRES in super-coiled form were obtained using the Ultra-pure Mini-plasmid Rapid Isolation Kit (Tiangen, Beijing, China), and their concentrations were detected by Nanodrop spectrophotometer (Thermo Scientific, Waltham, MA, USA). The transfection was performed using Lipofectamine™ LTX with Plus™ Reagent (Invitrogen, Carlsbad, CA, USA) following the manufacturer’s instructions. After transfection for 6 h, the medium was replaced with fresh growth medium. After transfection for 48 h, cells were collected for cyclic adenosine monophosphate (cAMP) assay and the culture medium was collected for hormone detection.

### 2.4. Expression of pIRES-S/CST14-S/2SS Plasmid in GH3 Cells

The expressions of SS and CST were identified by indirect immunofluorescence after transfection with the pIRES-S/CST14-S/2SS plasmid for 48 h as described in the previous study [19]. After washing, GH3 cells were fixed with 4% paraformaldehyde (PF) in PBS at 4 °C for 1 h. Then, the cells were permeabilized with 0.4% Triton X-100 for 20 min and blocked with 10% BSA for 30 min. Then, the cells were incubated with the anti-cortistatin (1:2000, Life Span BioSciences, Seattle, WA, USA) and anti-somatostatin (1:1000, Life Span BioSciences, Seattle, WA, USA) antibodies at 37 °C for 1 h. After washing three times with PBS, the cells were incubated at 37 °C for 1 h with the secondary antibodies, which were FITC labeled goat anti-rabbit antibodies (1:50, Boster, Wuhan, China). For negative control samples, primary antibodies were omitted and the same staining procedure was followed. For nuclear staining, cells were incubated with DAPI (4’,6-diamidino-2-phenylindole, 1:500, Sigma-Aldrich, Burlington, MA, USA). The results were analyzed by using a confocal laser scanning microscope (LSM 510 Meta instrument, Zeiss, Jena, Germany).

### 2.5. cAMP Assay after Transfection with Plasmids

GH3 pituitary cells were washed twice with 200 µL cold PBS after 48 h of transfection with plasmids, and then cAMP levels were measured by a cAMP measuring kit (Cyclic AMP XP Assay Kit #4339, Cell Signaling, Danvers, MA, USA) according to the manufacturer’s protocol. The data derived from the cAMP measurement was subjected to a nonlinear regression analysis.

### 2.6. Mice Immunization

A total of 108 three-week-old female Kunming mice were purchased from the Medical Laboratory Animal Center of Hubei Province (Wuhan, China) and housed in cages (five mice per cage) in an air-conditioned room at a constant temperature of 25 ± 2 °C with 12 h light/dark cycles and were provided with water and food ad libitum. The guidelines of the Committee on the Care and Use of Laboratory Animals of Huazhong Agricultural University were followed. The animals were divided into six groups (*n* = 18 in each group) according to the initial weight and a primary intramuscular immunization was given with 100 μL of either pIRES-S/CST14-S/2SS in 50 μg, 20 μg, 10 μg, pIRES-S/CST14 in 20 μg, pIRES-S/2SS in 20 μg or saline solution (negative control), respectively. A booster immunization was conducted two weeks later. Body weights were measured each week after primary immunization. Blood samples were harvested from the tail vein of each mouse at 2, 4, and 8 weeks after the booster immunization. Following centrifugation at 4000 rpm for 5 min, the plasma was separated from the whole blood and stored at −20 °C for further use.

### 2.7. Detection of Hormones and Antibodies

Growth hormone (GH) and the prolactin (PRL) concentrations in GH3 cells were measured by enzyme linked immunosorbent assay (ELISA) methods using commercial kits purchased from Wuhan Colorful Gene Biological Technology Co. LTD (Wuhan, China). The assay sensitivity of GH and PRL were 1.0 and 0.5 pg/mL, respectively. In addition, the GH and cortistatin concentrations in the mice plasma were measured by ^125^I labeled GH radio immunoassay and ^125^I labeled cortistatin radio immunoassay kits (Sinoukbio, Beijing, China) according to the manufacturer’s protocol. Somatostatin antibodies were measured as previously described by our lab [7].

### 2.8. Statistical Analysis

There were at least three replicates for each treatment. The data are presented as mean ± SEM. Data analysis was performed using GraphPad Prism software. A comparison of the mean values among groups was carried out using one-way ANOVA followed by either Tukey’s post hoc test or LSD test. *P* values of less than 0.05 and 0.01 were considered significant and extremely significant differences, respectively.

## 3. Results

### 3.1. Construction and Characterization of pIRES-S/CST14-S/2SS Plasmid

The electrophorogram for pIRES- S/CST14 -S/2SS showed that two bands with lengths of 717 bp and 6880 bp by *Nhe* I and *EcoR* I digestion, 789 bp and 6808 bp segments by *Sal* I and *Not* I digestion, and 2200 bp and 5397 bp fragments by *Nhe* I and *Not* I digestion were obtained, respectively (Figure 2I). The serial results of digestion and sequencing (data not shown) indicated that the construction of recombinant dual expression plasmid pIRES- S/CST14-S/2SS was successful. After transfection with the pIRES-S/CST14 -S/2SS plasmid for 48 h, green fluorescence signals were observed in transfected groups, suggesting both CST and SS can be expressed in GH3 cells by using IRES elements (Figure 2II).

### 3.2. The Biological Effect of Dual Expression Plasmid on Hormone Secretion and cAMP Accumulation

The levels of GH and PRL hormone in GH3 cells were examined after transfection with pIRES-S/CST14-S/2SS for 48 h. The results showed that both GH and PRL hormone levels significantly decreased when compared to the control group (*p* < 0.05, Figure 3I,II). Similarly, pIRES-S/CST14 transfected group observed the same tendency in reducing GH and PRL levels as the pIRES-S/CST14-S/2SS transfected group. However, there were no differences between the pIRES-S/2SS treated group and control group for GH secretion, indicating CST played a more pronounced role than SS at stimulating GH in GH3 cells. The changes for hormone production are mainly mediated by the second messenger cAMP. Therefore, we detected the cAMP levels after the transfection of different plasmids for 48 h. The results showed that the pIRES-S/CST14 transfected group exhibited significantly higher levels of cAMP than the pIRES-S/2SS transfected group and control group (*p* < 0.05). However, no significant difference in cAMP levels was observed between the dual expression pIRES-S/CST14-S/2SS transfected group and other groups (Figure 3III).

### 3.3. Immunization Effect of pIRES-S/CST14-S/2SS Plasmid on Hormone Levels and Growth in Mice

After immunization with different types of vaccines, anti-SS antibody in the plasma was measured. The results showed that the anti-SS antibodies were detected in all immunized groups, and the SS antibody levels in pIRES-S/CST14-S/2SS at a concentration of 10 μg was significantly higher than that in 50 μg of the pIRES-S/CST14-S/2SS group at week 4 after a booster immunization (*p* < 0.05, Figure 4I). However, no significant difference was observed among all vaccinated groups at week 2 and week 8 after the second immunization. We subsequently detected the levels of cortistatin and GH hormone in the plasma and found that pIRES-S/CST14-S/2SS at 10 μg and 50 μg showed lower levels of CST when compared to the control group (*p* < 0.01, Figure 4II). Moreover, the higher levels of GH were observed in all experimental groups rather than the control group (*p* < 0.05, Figure 4III).

In order to determine whether the changes in antibodies and GH production coupled to the increment of body weight, we further recorded the body weight and observed that mice immunized with 20 μg of pIRES-S/2SS showed a higher body weight at two weeks post-first immunization when compared to the pIRES-S/CST14 group ((*p* < 0.05, Table 1). Interestingly, mice in the 10 μg of pIRES-S/CST14-S/2SS group obtained the maximum weight gain and showed a significant difference in comparison to the control group (*p* < 0.05, Table 1), indicating that the pIRES-S/CST14-S/2SS vaccine has a dose-independent pattern.

## 4. Discussion

Although the previous studies have reported various somatostatin genetic vaccines including naked pCS/2SS plasmid and pVGS/2SS plasmid delivered by attenuated *Salmonella*, the growth-promoting effect of DNA immunization is still unsatisfied [7,11]. Cortistatin (CST) is a neuropeptide that shares high structural and functional homology with somatostatin and has a compensatory effect with somatostatin as found in single SS knockout mice. Simultaneous deletion of SS and CST mice exhibit a drastic elevation of endogenous GH levels, while IGF-1 levels and growth rate didn’t change [17]. Based on the above-mentioned observations, we hypothesized that cortistatin exerts the compensatory effect in single SS genetic immunization. Here, we developed recombinant dual expression CST-SS plasmids and hypothesized that it will improve the GH secretion and growth rate. To the best of our knowledge, it’s the first time to use an IRES dual expression system to study the compensatory effect of SS and CST. In the current study, a synthesized fragment of CST14 was fused with an HBsAg(S) gene, which contributes to the development of immunity against a variety of target gene products [20], and then both the S/CST and S/2SS fragments described in the previous report [7] were inserted into MCS-A and MCS-B sites of the pIRES plasmid, respectively, in order to produce the dual expression plasmid pIRES-S/CST14-S/2SS.

The biological functions of this novel plasmid were subsequently determined by detecting the secretion of GH and PRL in GH3 cells, which can express all five SSTRs [21]. After transfection with various plasmids, both the pIRES-S/CST14-S/2SS and pIRES-S/CST14 transfected groups had significantly lower levels of GH and PRL than the control group, while the single SS plasmid transfected group only displayed the significant suppressive effect on PRL secretion rather than GH levels, suggesting that cortistatin plays a pronounced role in the regulation of GH secretion. It has been well documented that cortistatin can similarly bind to the five native somatostatin receptors, resulting in the inhibitory effect on GH and PRL secretion [14]. Interestingly, SS caused lower cAMP accumulation, which mediates many inhibitory functions of the somatostatin family [22], whereas CST stimulated higher cAMP levels, indicating the opposite effect in terms of cAMP levels. The possible explanation is due to the specific binding activity of CST rather than SS to the ghrelin receptor (GHSR) [23], which can activate the cAMP/PKA signaling pathway associated with anti-inflammatory reactions [24].

After the injection of mice with the CST and SS dual expression plasmid, we found that all does of pIRES-S/CST14-S/2SS immunized groups exhibited higher levels of GH hormone compared to the control group, which was consistent with the previous studies in SS and CST DKO mice [17]. We also observed a similar growth trend in all investigated groups during the experimental period. The group at a low dose (10 μg) of pIRES-S/CST14-S/2SS plasmid showed maximum body weight gains during the second injection period, However, no significant difference in weight gains was observed either in the high dose group (50 μg) or medium dose group (20 μg), indicating the growth-promoting effect of this vaccine is dose-independent. Luque has demonstrated that both somatostatin and cortistatin display a dual [25], dose-dependent stimulatory and inhibitory effect on GH secretion in somatotropes, which reminds us that the dose is a pertinently important factor in the progress of SS and CST vaccine development. It should be noted that single pIRES-S/CST14 and pIRES-S/2SS also presented an anticipatory effect on the growth improvement in mice but required a higher concentration.

## 5. Conclusions

It can be concluded that the novel dual expression pIRES-S/CST14-S/2SS plasmid was successfully constructed and could induce higher GH levels and enhance weight gain in mice in a dose-independent manner. This study helps us to better understand the relationship between somatostatin and cortistatin, and lays a foundation for the development of a growth-promoting DNA vaccine.

## Figures and Tables

**Figure 1 animals-12-01490-f001:**
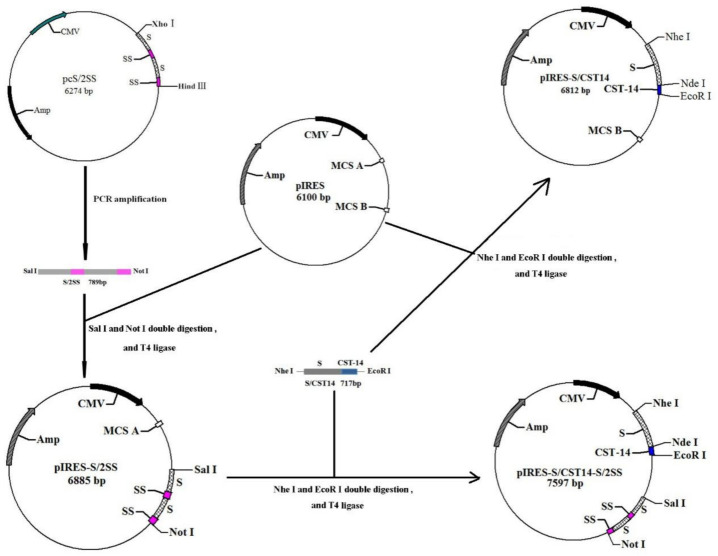
Schematic diagram for the construction of plasmid pIRES-S/CST14 -S/2SS. The S/2SS gene encoding two copies of somatostatin genes presented by the hepatitis B surface antigen (HBsAg) particle was amplified by PCR with *Sal* I and *Not*I. After purification, the target gene was inserted into the pIRES vector. Corstitatin-14 was chemically synthesized and fused with the hepatitis B surface antigen (HBsAg) genes, which formed the S/CST14 fragment with the *Nhe* I and *EcoR* I sites. The S/CST14 fragment was then inserted into pIRES-S/2SS plasmid to construct the dual expression plasmid pIRES-S/CST14-S/2SS.

**Figure 2 animals-12-01490-f002:**
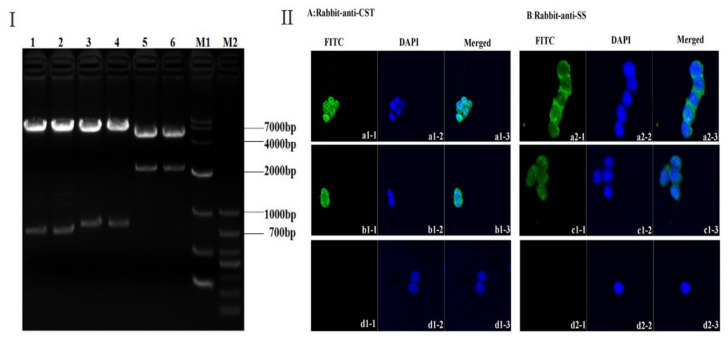
(**I**) Agarose gel electrophoresis of pIRES-S/CST14-S/2SS plasmid digested with restriction endonucleases. Lanes 1 and 2 showed two bands of 717 bp and 6880 bp with *Nhe* I and *EcoR* I digestion; Lanes 3 and 4 showed two bands of 789 bp and 6808 bp with *Sal* I and *Not* I digestion; and Lanes 5 and 6 showed two bands of 2200 bp and 5397 bp with *Nhe* I and *Not* I digestion. (**II**) Confirmation of expression of S/CST14 and S/2SS fusion protein in pIRES-S/CST14-S/2SS (a), pIRES-S/CST14 (b), pIRES-S/2SS (c), and pIRES (d, control) transfected GH3 cells, respectively. Images were captured using a confocal microscope. The S/CST14 fusion protein could be clearly observed in GH3 cells (a1-1, a1-3, b1-1, and b1-3). Also, the S/2SS fusion protein could be clearly observed in GH3 cells (a2-1, a2-3, c1-1, and c1-3).

**Figure 3 animals-12-01490-f003:**
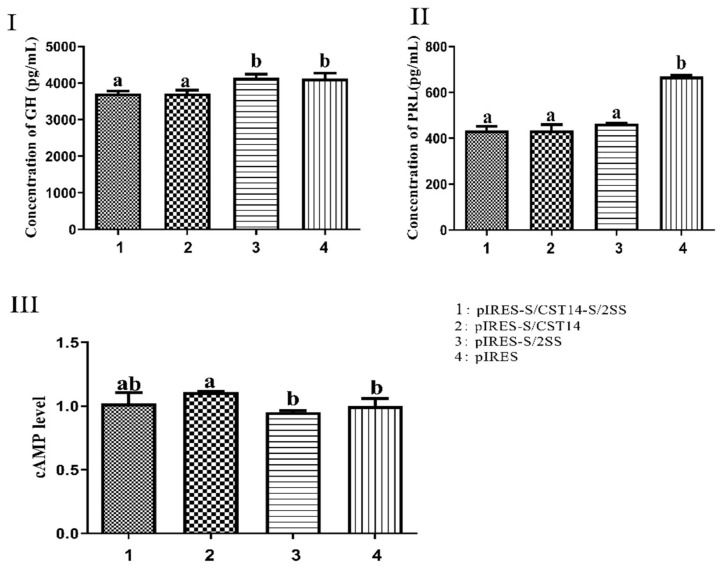
(**I**,**II**) The GH and PRL concentrations in GH3 cells transfected with four various plasmids for 48 h. (**III**) The intracellular cAMP accumulation in GH3 cells transfected with four various plasmids for 48 h. The results are expressed as the mean ± SEM. The same letters on the bar represent no significant difference, different letters on the bar represent significant difference at *p* < 0.05.

**Figure 4 animals-12-01490-f004:**
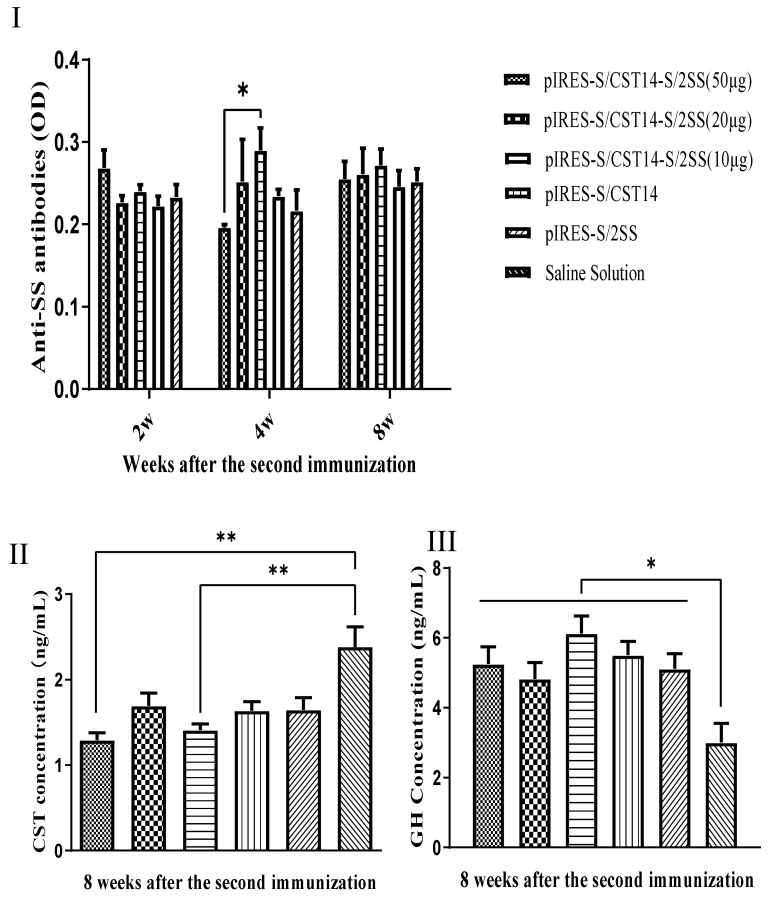
(**I**) The detection of anti-somatostatin antibodies in mice immunization with 100 μL of pIRES-S/CST14 -S/2SS (50 μg), pIRES-S/CST14 -S/2SS (20 μg), pIRES-S/CST14-S/2SS (10 μg), pIRES-S/CST14 (20 μg), pIRES-S/2SS (20 μg), and saline solution at 2, 4, and 8 weeks after a booster immunization. (**II**,**III**) Cortistatin and growth hormone concentration of all groups were measured at 8 weeks after the second vaccination. The results are expressed as the mean ± SEM. * means significant differences at *p* < 0.05, and ** means significant differences at *p* < 0.01.

**Table 1 animals-12-01490-t001:** Weight gain of mice immunized with various plasmids (mean ± SEM, g).

Group	Initial Weight	Weight of 2 w after the First Immunization	Final Weight	Total Weight Gain of the Second Immunization
pIRES-S/CST14-S/2SS (50 μg)	17.4 ± 0.12	27.6 ± 0.26 ^ab^	36.1 ±0.31	8.5 ±0.19 ^a^
pIRES-S/CST14-S/2SS (20 μg)	17.3 ± 0.20	26.5 ± 0.32 ^ab^	35.4 ±0.38	8.9 ±0.23 ^ab^
pIRES-S/CST14-S/2SS (10 μg)	17.6 ± 0.13	27.3 ± 0.21 ^ab^	38.1 ±0.25	10.8 ±0.16 ^b^
pIRES-S/CST14	17.3 ± 0.10	25.5 ± 0.20 ^a^	36.1 ±0.35	10.6 ±0.23 ^b^
pIRES-S/2SS	18.0 ± 0.11	27.7 ± 0.17 ^b^	37.3 ±0.28	9.6 ±0.18 ^ab^
Saline Solution	17.7 ± 0.14	26.8 ± 0.20 ^ab^	35.4 ±0.28	8.6 ±0.16 ^a^

Note. Same letters within a column represent no significant difference, and different letters within a column represent significant difference at *p* < 0.05.

## Data Availability

The data that support the results of this study are available from the corresponding author upon reasonable request.

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
