# Peer review of "Evaluation of a Novel DNA Vaccine Double Encoding Somatostatin and Cortistatin for Promoting the Growth of Mice"

_animals, 2022, doi:10.3390/ani12121490_

Round 1

Reviewer 1 Report

  1. This paper was well organized and written at the beginning of the manuscript, including the abstract and introduction, however, in the material section and results, lots of grammar problems need to be improved. Even a lot of typos in the manuscript, for example, in line 104, “All cells were cultured at 37 in an incubator ……”. Please recheck the language editing for the whole manuscript in the revision. This is the author’s responsibility.
  2. Some background issues need to be described in this paper, including: Why use HBV surface antigen as adjuvant protein? Please describe it. Why use pIRES for antigen expression purposes? Could the expression of two antigens be varying after transfection?
  3. For Figure 2-II., please describe the staining details, for example: did the cell be fixed before antibody probing. Please provide the cell morphology by the bright field image. The cell density seems quite low, which does not correspond with the 70~80 confluence at the transfection described in the method section.
  4. In results section 3.2, have any rationale that can explain why do this experiment? It’s too difficult to understand, why an in-vitro cultured cell GH expression can be inhibited by a DNA vaccine? Is it a kind of shRNA? This is not consistent with common sense, please describe a possible mechanism in detail.

Author Response

Following the reviewers’ helpful comments, we have carefully revised our manuscript entitled “Evaluation of a novel DNA vaccine double encoding somatostatin and cortistatin for promoting the growth of mice” (Manuscript ID: animals-1678252). We are now submitting the revised manuscript along with a letter outlining the changes in response to the reviewers’ comments, please see the attachment.

Reviewer 2 Report

  1. There is need for minor grammar and english words revision
  2. Author emphasised large animals in objectives but did not do any experiment on large animal thus either it should be removed or experiment on large animal may please be included 
  3. In introduction section such work on large animals should be reviewed 
  4. Self citation should be minimum
  5. Plagiarism level should be reduced

Author Response

Following your helpful comments, we have carefully revised our manuscript entitled “Evaluation of a novel DNA vaccine double encoding somatostatin and cortistatin for promoting the growth of mice” (Manuscript ID: animals-1678252). We are now submitting the revised manuscript along with a letter outlining the changes in response to your comments , please see the attchment.

Reviewer 3 Report

The article “Evaluation of a novel DNA vaccine double encoding somatostatin and cortistatin for promoting the growth of mice” here proposed is very well presented, increases the understanding of  correlation between somatostatin and cortistatin, and may lays a foundation for development of a growth promoting DNA vaccine. . I consider that this paper fits into scope of this scientific journal  and I would definitely recommend this paper for publication after the revision of the following points:

Line 55: Put the initial letter of “somatostin” in lowercase

Line 84: Divide the period by inserting a period between “Not I” and “after”.

Materials and methods: 2.6 Mice immunization: please, specify better the sample sice of each group

Line 153: what is 125I-GH? I don’t find the explanation of this aabbreviation.

Line 155 - 156:  In my opinion it would be better “ The GH concentration in plasma mice…”

Line 181: I don’t find the explanation of  “DAPI” abbreviation.

Line 239: Please insert citations.

References:

Lines 315, 351,352,and 359: All initials are written in capital letters. Please correct.

Line 321 and Line 338: Please, write “Salmonella typhimurium” in italics

Line 345: Please, write “2017” in bold type

Author Response

Following your helpful comments, we have carefully revised our manuscript entitled “Evaluation of a novel DNA vaccine double encoding somatostatin and cortistatin for promoting the growth of mice” (Manuscript ID: animals-1678252). We are now submitting the revised manuscript along with a letter outlining the changes in response to your comments, please see the attachment.

Round 2

Reviewer 1 Report

Thanks for the author's responses, the comment points are all satisfactory.